# Automated Segmentation of Knee Bone and Cartilage combining Statistical Shape Knowledge and Convolutional Neural Networks

## Data from the Osteoarthritis Initiative

**Felix Ambellan**$^{a,*}$, **Alexander Tack**$^{a,*}$, Moritz Ehlke$^{b,a}$, Stefan Zachow$^{a,b}$

$^a$Zuse Institute Berlin, $^b$1000shapes GmbH, Berlin, Germany

{ambellan, tack, ehlke, zachow}@zib.de

## Abstract

We present a method for the automated segmentation of knee bones and cartilage from magnetic resonance imaging, that combines *a priori* knowledge of anatomical shape with Convolutional Neural Networks (CNNs). The proposed approach incorporates 3D Statistical Shape Models (SSMs) as well as 2D and 3D CNNs to achieve a robust and accurate segmentation of even highly pathological knee structures. The method is evaluated on data of the MICCAI grand challenge "Segmentation of Knee Images 2010". For the first time an accuracy equivalent to the inter-observer variability of human readers has been achieved in this challenge. Moreover, the quality of the proposed method is thoroughly assessed using various measures for 507 manual segmentations of bone and cartilage, and 88 additional manual segmentations of cartilage. Our method yields sub-voxel accuracy. In conclusion, combining of anatomical knowledge using SSMs with localized classification via CNNs results in a state-of-the-art segmentation method.

## 1 Introduction

Knee osteoarthritis (OA) is a chronic, degenerative joint disease affecting a significant fraction of the human population [1]. Due to the rising average life expectancy, an increasing obesity and interest in an active lifestyle, research to understand and prevent OA will become even more important. Magnetic Resonance Imaging (MRI) is commonly used to assess knee joint degeneration, especially of the femoral bone (FB), tibial bone (TB), and the respective femoral and tibial cartilage (FC,TC). Quantitative image-based biomarkers from MRI, e.g. the apparent bone volume divided by total bone tissue volume [2] or the cartilage volume [3], already show potential for diagnosis of OA, treatment planning, and prognostic purposes. However, the determination of such biomarkers requires the segmentation of bone and/or cartilage.

Clinical studies with a large number of subjects are required to extract quantitative image-based biomarkers indicating pathogenesis of OA and to evaluate the efficacy of therapeutic approaches. Precise segmentations are also a prerequisite for computer-based surgical planning of interventions affecting the knee. Manual segmentation of the knee joint is, however, tedious, subjective, and labor-intensive, which renders the analysis of larger cohorts impractical. Thus, since many years, the performance and quality of automated methods have been improved and new methods are being developed.

Earlier methods often employed Statistical Shape Models (SSMs) [4] to segment knee bones and cartilage. In 2010, Vincent et al. [5] presented a method based on an active appearance model, which was created using a minimum description length approach to optimize correspondences. In the same

---

$^*$Equal contribution

1st Conference on Medical Imaging with Deep Learning (MIDL 2018), Amsterdam, The Netherlands.

year, Seim et al. [6] presented a method that utilizes SSMs for bone segmentation and a multi-object graph optimization for cartilage segmentation. SSM-based methods employ anatomical knowledge via geometric priors, which allows for a robust segmentation even in the presence of artifacts or low image contrast. Such methods usually require heuristically designed models of appearance, to adjust the SSM to the image data. Often, appearance models are manually fine-tuned to one specific image modality and cannot be easily generalized to cope with differing ones. To alleviate this problem, Mukhopadhyay et al. [7] proposed to derive appearance models directly from the image data via joint dictionary learning.

Meanwhile, Convolutional Neural Networks (CNNs) have been employed successfully for segmentation tasks on medical image data, but only few of these methods address the domain of musculoskeletal research. In 2013, Prasoon et al. [8] presented an approach for tibial cartilage segmentation from MRIs using three 2D CNNs. Each CNN independently classifies foreground and background pixels from slices in either the axial, coronal, and sagittal image planes of the 3D MRI dataset. In a similar fashion, Liu et al. [9] in 2017 applied 2D U-Nets [10] as well as the 2D CNN architecture "SegNet" in combination with 3D simplex deformable modeling to obtain 3D segmentations from MRIs. Both methods train 2D convolutional filters from individual slices in the 3D MRI data, since the memory consumption of deep 3D CNNs is often excessive at the scale of full-resolution 3D medical datasets. Consequently, the image information available to the CNNs is strictly localized and lacks context w.r.t. the surrounding voxel intensities in neighboring slices. This is in contrast to previous SSM-based approaches, where 3D anatomical shape information regularizes the segmentation outcome across several slices.

Our aim in this work is to improve on the segmentation accuracy of previous approaches by combining the strengths of SSM-based and CNN-based segmentation methods. Another aim is to segment large cohort data automatically, e.g. the databases of the Osteoarthritis Initiative[2] (OAI) or the Study of Health in Pomerania[3]. We propose a method that utilizes 2D CNNs as well as localized 3D CNNs to incorporate as much context information as possible into the segmentation process. SSMs are integrated into our segmentation workflow to support decision making in areas of low confidence through a voting scheme. The segmentation accuracy is validated based on a large pool of diverse datasets from the MICCAI "Segmentation of Knee Images 2010"[4] [11] (SKI10) challenge and the OAI, as described in section 3. By utilizing SSMs as anatomical shape prior for regularization and CNNs for learning descriptors of local appearance, our method robustly segments varying MRI sequences, even when the images show subjects with severe OA grades.

## 2 Automated segmentation of bone and cartilage

Our aim is to establish an automated method that produces highly accurate segmentations of the knee and is robust against pathological data, imaging artifacts, as well as the varying image appearance in different MRI sequences. For this purpose, we consecutively apply 2D CNNs, sub-regional 3D CNNs and SSM-based regularization for both steps, yielding accurate bone segmentations. Given the margins of the bones, we then extract subvolumes along the femoral condyles and tibial plateaus and segment the cartilage in these regions using 3D CNNs (cf. Fig. 1). The first step *CNN-2D* creates initial segmentation masks of FB and TB. The second step *SSM adjustment* regularizes the results of step *CNN-2D* by fitting SSMs to these masks. The third step *CNN-3D* is a refinement step that employs 3D CNNs to segment small MRI subvolumes at the bone surfaces as given by the preceding *SSM adjustment*. The fourth step *SSM postprocessing* uses regions pre-defined on SSMs to regularize the results of *CNN-3D*. After bone segmentation is finished the FC and TC are segmented using 3D CNNs. Each step is performed separately for femur and tibia. Thus, CNNs and SSMs are developed independently and individually for both structures. In the following, details are given for all steps of our segmentation pipeline.

### 2.1 CNN-2D

The first step *CNN-2D* is inspired by Liu et al. [9] and applies a variant of the 2D U-Net (Fig. 2 left) for the independent, slice-wise segmentation of the image data. Training is carried out using the

---

[2]https://oai.epi-ucsf.org
[3]http://www2.medizin.uni-greifswald.de/cm/fv/ship.html
[4]organized by Tobias Heimann and Bram van Ginneken (http://ski10.org)

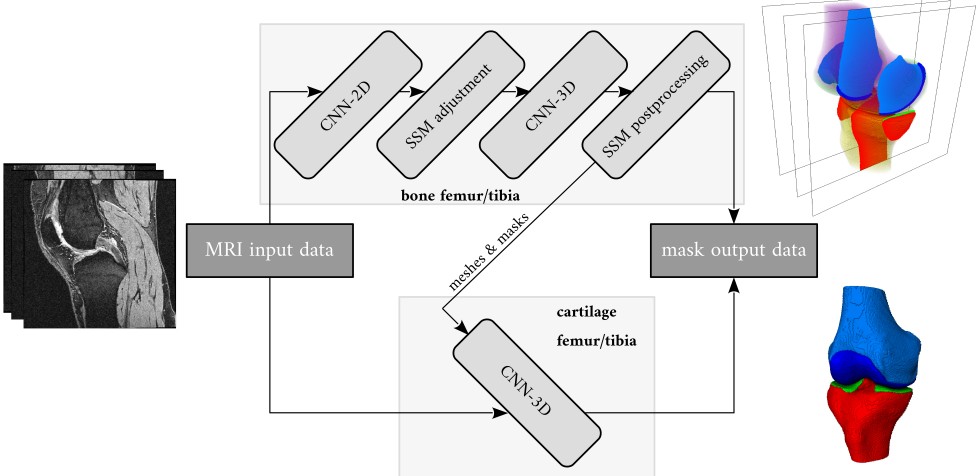

Figure 1: Proposed pipeline for knee bone and cartilage segmentation.

(slice-wise) DICE similarity coefficient (cf. Appendix A) as a loss function. The number of input channels of the 2D CNN was extended compared to the standard architecture in order to improve spatial consistency of segmentation results between individual slices of the MRI stack. Eight adjacent slices on both sides of the one that is to be segmented are additionally supplied resulting in 17 channels in total. Note that while this method provides additional context information to the CNN, it cannot substitute for the true volumetric input as processed by 3D CNNs, since the additional channels are only directly visible to the first convolutional layer. The memory requirements are, however, significantly reduced compared to 3D U-Nets with a similar architecture.

## 2.2 SSM adjustment

Due to our observation that segmentations of the previous step show inaccuracies in areas of low intensity contrast or imaging artifacts (cf. Fig. 3 left column and Fig. 4 upper left column) we decided to add a regularization step. The aim of the *SSM adjustment* step is to regularize and to fill holes and notches in the segmentation mask through statistical knowledge about the global variation of anatomical shape. For this purpose, an SSM (cf. Section 3) is fitted to the segmentation results from the *CNN-2D* stage. The output is guaranteed to be anatomically plausible (i.e. within the shape span of the SSM) and given as one connected component. Further details on SSMs, their construction and adjustment can be found in [12].

The SSM matching procedure is as follows: In order to initialize the SSM considering the side of the knee (left/right) a template mesh of the condyle region is fitted to the mask, one time affinely and a second time additionally mirrored along the epi-condyle axis. The actual knee side is detected according to the lower distance between template and data. The shape modes of the SSM and similarity transformation are adjusted iteratively to fit the vertex positions of the SSM to the mask:

$$\underset{v\left(b^{i+1}, T^{i+1}\right)}{\arg\min} \left\| \left( v\left(b^i, T^i\right) + \Delta v^i \right) - v\left(b^{i+1}, T^{i+1}\right) \right\|, \;\; i \leftarrow i + 1, \tag{1}$$

where $\Delta v^i$ is the displacement along the normals of the vertices resulting from the $i$-th step, s.t. they are placed as close as possible to the interface between segmentation mask (intensity value = 1) and background (intensity value = 0). The vertex positions, obtained from the SSM w.r.t. shape weights $b^i$ and transformation $T^i$, are denoted by $v\left(b^i, T^i\right)$.

## 2.3 CNN-3D

SSMs, as utilized in the previous stage of the pipeline, cannot express osteophytic details completely since these deformations are highly patient-specific and might not be derived from the training cohort.

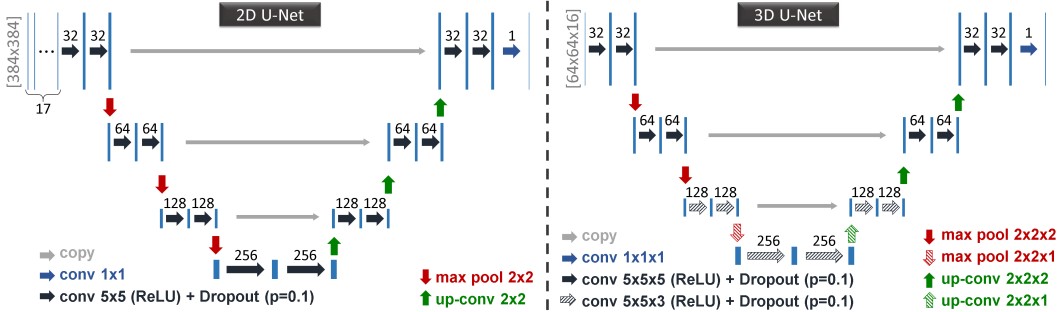

Figure 2: Architectures of the 2D and 3D U-Nets employed in this work.

We approach this issue in the *CNN-3D* step by employing 3D U-Nets (Fig. 2 right) with local input fields to segment MRI subvolumes along the bone contours. This step is carried out utilizing the same architecture, but individual training for every structure to capture anatomical details of bone tissue as well as cartilage. Similar to the 2D-U-Nets, the networks are trained via a loss function based on the DICE similarity coefficient. The loss is, however, defined on volumetric subvolumes in the MRI rather than individual 2D slices, which generally leads to better classification accuracy in local regions of the knee compared to the 2D slicing approach. We set the dimensions of the subvolumes to 64x64x16 voxels, compromising between the input fields size and the memory consumption of the 3D CNNs. To get a feasible number ($\approx 1000$) of subvolumes they are randomly sampled following a Poisson distribution for the FB and TB segmentation along the outline of the adjusted SSM's mask. The subvolumes for FC segmentation are extracted along the outline of the condyle region in a similar manner. For TC segmentation, the natural (almost) planar conditions of the tibial plateaus are utilized and subvolumes are sampled at the superior margin of each sagittal slice in the TB mask, s.t. the in-plane overlap of subvolumes is half its size. A visualization is given in Appendix C.

Since the subvolumes are partially overlapping, conflicting labels might be assigned in overlapping regions. A voting scheme solves this ambiguity by summarizing voxel-wise decisions in a voting mask $V$ (2). The outcome of the *SSM adjustment* stage ($G$) is thereby taken into account, biasing the FB and TB segmentations towards the SSM shape in case votes conflict. The contribution of the SSM is set to zero for the segmentation of cartilage.

Voting can be formularized as follows: Let $\mathbf{x} = (i, j, k)$ be a (global) index triplet of an image voxel. Let further be $I_s$ the mapping from local indices of subvolume $B_s$ to global indices in the image, and let $\mathrm{Im}(I_s)$ be the set of all global indices covered by $B_s$. The voting mask is computed as:

$$V(\mathbf{x}) = \omega \cdot G(\mathbf{x}) + \sum_{\substack{s \\ \mathbf{x} \in \mathrm{Im}(I_s)}} \Big( 2 \cdot B_s\big(I_s^{-1}(\mathbf{x})\big) - 1 \Big), \tag{2}$$

with factor $\omega$ set to 25, weighting the contribution of SSM and CNNs roughly equal. The higher $\omega$ is chosen, the more trust is put into the SSM-based regularization. Using this voting mask, majority voting is applied to generate the segmentation mask $U$:

$$U(\mathbf{x}) = \left\{ \begin{array}{ll} 1, & \text{if } V(\mathbf{x}) \geq \tau \\ 0, & \text{if } V(\mathbf{x}) < \tau, \end{array} \right.$$

where the threshold parameter $\tau$ has a fixed value of 1, since this way, a majority of CNN-classified subvolumes have to up-vote a voxel in order to include it in the final result, if it is not already captured by the SSM.

## 2.4 SSM postprocessing

Bone segmentation is finalized in the *SSM postprocessing* step. The idea behind SSM-based postprocessing as utilized in our approach is to remove wrongly classified foreground labels from the FB and TB segmentations that are located outside the typical range of osteophytic growth. Foreground labels are excluded from the segmentation mask depending on their surface distance to the SSM. We found that a conservative regularization after processing the subvolumes with the proposed voting

method (*CNN-3D* stage) helps to remove segmentation errors due to the localized nature of the 3D CNN-based classification and low image contrast or noise (cf. Fig. 4, central and right column).

We identified regions on the SSMs of the FB and TB that show higher or lower inter-patient variability in shape (cf. Appendix B). Regions of higher variability are typically associated with osteophytic growth. To identify these regions in the segmentation, SSMs are matched to the masks obtained from the *CNN-3D* stage and the distance between foreground voxels and fitted surface is calculated, s.t. every voxel is assigned a distance value either to areas of low or high variability on the SSMs. This leads to a maximum distance in areas of low variability $d_{lv}$ and to a maximum distance in areas of high variability $d_{hv}$.

If $d_{hv} \geq d_{lv}$, the postprocessing is terminated since no unexpected deviation is identified. Otherwise, the matched surface is converted to a mask denoted by $\tilde{U}$ and the set difference $D = U \setminus \tilde{U}$ is calculated. D naturally consists of its 3D connected components $D_j$. Every component is considered a candidate for removal, if there exists a voxel $x \in D_j$, s.t. its distance to the surface $d_{\mathbf{x}}$ is realized in the area of low variability and additionally $d_{\mathbf{x}} \geq d_{hv}$ and $d_{\mathbf{x}} \geq tol$ hold for an empirically determined tolerance of 5.5mm. All other components describe either osteophytic growth or normal morphological deviation. For components that fulfill the former rule, every voxel that realizes its distance in the area of low variability is removed from the mask. See Appendix B for a detailed schematic description.

## 3 Experiments and Results

We evaluated the accuracy of our method on three different datasets *SKI10*, *OAI Imorphics* and *OAI ZIB* (Table 1) employing volume-based and distance-based measures (cf. Appendix A).

### 3.1 MRI datasets

Dataset *SKI10* consists of MRIs from the MICCAI SKI10 grand challenge. These data is divided into 60 training, 40 validation, and 50 submission images. All scans were acquired for surgery planning of partial or complete knee replacement, and thus show a high degree of pathological deformities in the knee region.

Dataset *OAI Imorphics* consists of MRI sequences from the OAI database with manual segmentations supplied by Imorphics (N = 88). The dataset contains only cases of moderate and severe OA.

Dataset *OAI ZIB* consists of additional data from the OAI database for which manual segmentations were carried out thoroughly by experienced users at Zuse Institute Berlin (N = 507) starting from automatic segmentations employing [6]. The data cover the full spectrum of OA grades, with a strong tendency towards severe cases.

Table 1: Summary of the datasets used for training and validation. Images were acquired either once per patient (baseline) or twice with an additional 12-month follow-up (12m).

|  | SKI10 | OAI Imorphics | OAI ZIB |
|---|---|---|---|
| MRI scanner | GE, Siemens, Philips, Toshiba, Hitachi. Mostly 1.5T, some 3T, a few 1T | Siemens 3T Trio | Siemens 3T Trio |
| MRI sequence | Many (T1, T2, GRE, Spoiled-GRE) partly with fat suppression | DESS | DESS |
| Acquisition plane | sagittal | sagittal | sagittal |
| Image resolution [mm] | 0.39×0.39×1.0 | 0.36×0.36×0.7 | 0.36×0.36×0.7 |
| Manual segmentations | bones and cartilage | cartilage | bones and cartilage |
| Number of subjects | 60 training 40 validation 50 submission | 88 | 507 |
| Sex (male;female) | n.a. | (45,43) | (262,245) |
| Age [years] | n.a. | 61.24±9.98 | 61.87±9.33 |
| BMI [kg/m$^2$] | n.a. | 31.06±4.61 | 29.27±4.52 |
| rOA grade (0,1,2,3,4) | n.a. | (0,0,15,56,17) | (60,77,61,151,158) |
| timepoints | baseline | baseline, 12m | baseline |

### 3.2 Experimental setup

The employed SSMs consist of 15,172 vertices and 30,220 faces (FB), and 16,244 vertices and 32,351 faces (TB) independent of the dataset. Construction was done following [6]. For the *SKI10* dataset, training of CNNs and construction of SSMs is carried out using the 60 training cases. Our method is evaluated for the validation and the submission cases separately. Two-fold cross-validation studies are performed for datasets *OAI ZIB* and *OAI Imorphics*. For *OAI ZIB*, decomposition is done by random choice (253/254). For *OAI Imorphics* the cohort's subject ids are sorted numerically and split into upper and lower half (44/44). However, since no manual segmentations of bones are available for the *OAI Imorphics* dataset, the SSMs built from the *SKI10* training data are employed. The CNNs are trained using the *OAI Imorphics* baseline data only. Thus, the *OAI Imorphics* 12m follow-up data is exclusively used for evaluation still within the cross-validation setting.

### 3.3 Measures of segmentation accuracy

The accuracy of our method is evaluated using the DICE Similarity Coefficient (DSC), average surface distance (ASD), root mean square distance (RSD), maximum distance (MSD), volume difference (VD), and volume overlap error (VOE). All these measures are symmetric apart from VD that is considered relative to the manual segmentation. Volumetric measures (Appendix A, 3) are suitable for assessing the segmentation results globally. However, volume-based measures provide limited sensitivity to errors on the boundaries of the segmentation if the segmented volume is relatively large. We therefore also include surface distance measures (Appendix A, 4) in the evaluation, which are sensitive to segmentation errors on the anatomical boundary.

### 3.4 Results

Table 2 summarizes our results for the *SKI10* validation dataset. Our method reaches a total score of 73.6±7.6 in terms of the SKI10 metrics [11]. This is a notable improvement w.r.t scores reported in previous publications (Table 3). Our method achieves a total score of 75.73 on the *SKI10* submit data and is currently ranked first[5].

The results for the *OAI Imorphics* dataset are shown in Table 4. For FC the DSC is 89.4% for baseline and 89.1% for 12m. For medial tibial cartilage (MTC) and lateral tibial cartilage (LTC) the DSC is 86.1% resp. 90.4% for baseline, and 85.8% resp. 90.0% for 12m. Again, the ASD is smaller than the image resolution (<0.36mm) for both, FC and TC.

Table 5 summarizes the segmentation accuracy for the *OAI ZIB* dataset. The DSC is 98.5% for FB, 98.5% for TB, 89.9% for FC, and 85.6% for TC. The ASD is smaller than the image resolution for bone as well as for cartilage (<0.36mm).

Computation for the whole segmentation pipeline (end to end) was 9m 22s on a consumer-grade workstation (CPU: Intel Xeon E5-2650 v3, 2.30GHz; GPU: GeForce GTX 980 Ti). Implementation of CNNs was done employing Keras with Theano-backend[6]. All calculations regarding SSMs were carried out using *Amira ZIB Edition*[7]

Table 2: Segmentation accuracy for the *SKI10* validation dataset.

|        | ASD (mm)        | RSD (mm)        | VD (%)          | VOE (%)          |
|--------|-----------------|-----------------|-----------------|------------------|
| **FB** | $0.43 \pm 0.13$ | $0.75 \pm 0.28$ | —               | —                |
| **TB** | $0.37 \pm 0.11$ | $0.63 \pm 0.26$ | —               | —                |
| **FC** | —               | —               | $7.18 \pm 10.51$ | $20.99 \pm 5.08$ |
| **TC** | —               | —               | $4.29 \pm 12.34$ | $19.06 \pm 5.18$ |

Total score as computed employing the SKI10 metrics: $73.6 \pm 7.6$

## 4 Discussion and Conclusion

We presented a novel fully automated segmentation method for knee bone and cartilage by combining the advantages of SSM-based regularization with CNN-based classification of voxel intensities. The

---

[5]http://www.ski10.org/results.php, as of April 2018

[6]https://keras.io

[7]https://amira.zib.de/download.html

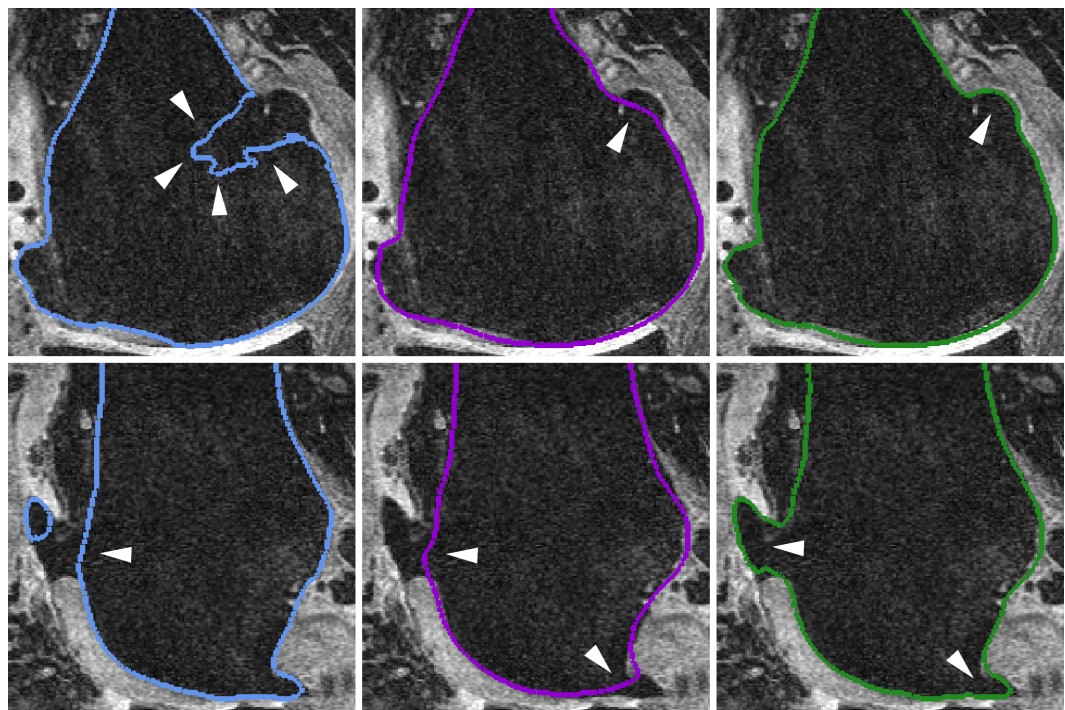

Figure 3: Segmentation of osteophytic regions in different stages (colored contours). *CNN-2D* stage (left) is error prone, *SSM adjustment* (middle) smoothly regularizes and *CNN-3D* (right) segments osteophytes precisely.

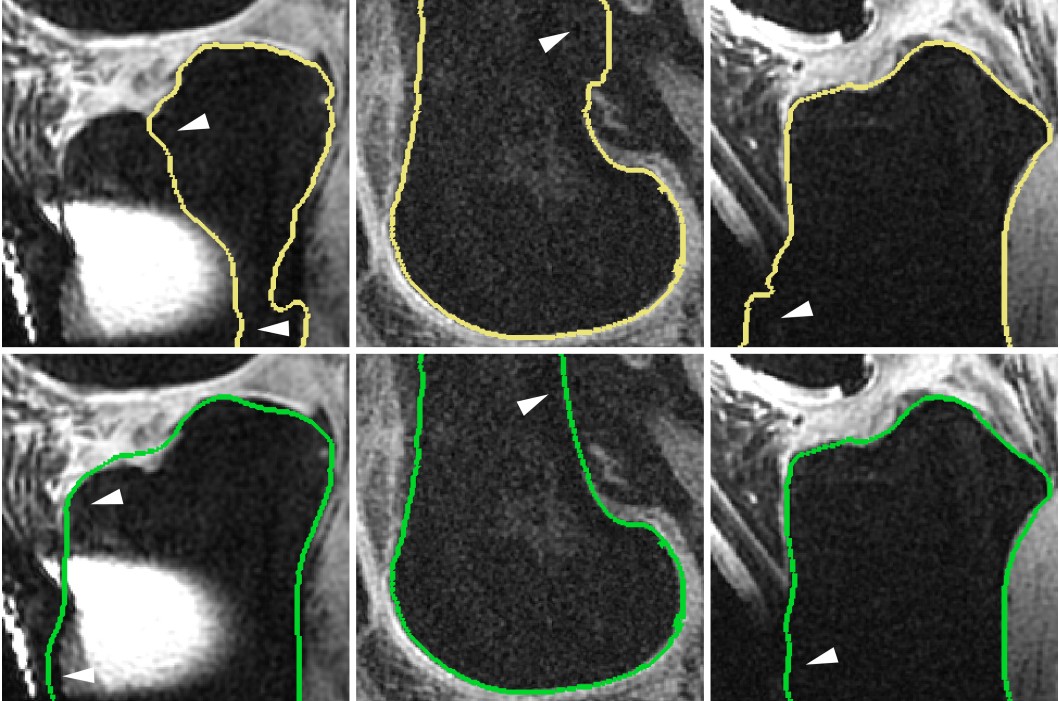

Figure 4: Left column: An image artifact results in an erroneous *CNN-2D* classification (top). An anatomically plausible segmentation (bottom) is restored using *SSM adjustment* regularization. Middle and right column: *CNN-3D* segmentation introduces errors in the shaft region due to the sub-volumes' locality and missing image contrast (top). *SSM postprocessing* corrects in an anatomically plausible manner (bottom).

Table 3: *SKI10* validation data: Our approach yields the best of all published results as of April 2018.

| Imorphics [5] | ZIB (2010) [6] | BioMedIA [13] | Liu et al. [9] | Biomediq [14] | **ZIB (2018)** |
|---|---|---|---|---|---|
| $52.3 \pm 8.6$ | $54.4 \pm 8.8$ | $56.5 \pm 9.2$ | $64.1 \pm 9.5$ | $67.1 \pm 8.0$ | **$73.6 \pm 7.6$** |

Table 4: Segmentation accuracy for the *OAI Imorphics* dataset.

| | | DSC (%) | ASD (mm) | RSD (mm) | MSD (mm) | VD (%) | VOE (%) |
|---|---|---|---|---|---|---|---|
| 00m | **FC** | $89.4 \pm 2.41$ | $0.19 \pm 0.08$ | $0.50 \pm 0.30$ | $6.65 \pm 2.99$ | $7.0 \pm 6.04$ | $19.1 \pm 3.88$ |
| | **MTC** | $86.1 \pm 5.33$ | $0.26 \pm 0.23$ | $0.63 \pm 0.55$ | $5.16 \pm 2.93$ | $8.0 \pm 17.15$ | $24.1 \pm 7.74$ |
| | **LTC** | $90.4 \pm 2.42$ | $0.17 \pm 0.06$ | $0.41 \pm 0.16$ | $3.93 \pm 2.07$ | $6.9 \pm 7.14$ | $17.5 \pm 3.96$ |
| 12m | **FC** | $89.1 \pm 2.41$ | $0.20 \pm 0.09$ | $0.53 \pm 0.33$ | $6.86 \pm 3.16$ | $7.6 \pm 6.78$ | $19.6 \pm 3.86$ |
| | **MTC** | $85.8 \pm 5.00$ | $0.28 \pm 0.22$ | $0.67 \pm 0.55$ | $5.25 \pm 3.06$ | $6.6 \pm 16.38$ | $24.5 \pm 7.37$ |
| | **LTC** | $90.0 \pm 2.57$ | $0.18 \pm 0.06$ | $0.44 \pm 0.19$ | $4.08 \pm 2.11$ | $7.2 \pm 7.74$ | $18.1 \pm 4.16$ |

Table 5: Segmentation accuracy on the *OAI ZIB* dataset.

| | DSC (%) | ASD (mm) | RSD (mm) | MSD (mm) | VD (%) | VOE (%) |
|---|---|---|---|---|---|---|
| **FB** | $98.5 \pm 3.02$ | $0.17 \pm 0.05$ | $0.35 \pm 0.09$ | $2.93 \pm 1.24$ | $-0.09 \pm 0.87$ | $2.8 \pm 0.58$ |
| **TB** | $98.5 \pm 3.25$ | $0.18 \pm 0.06$ | $0.37 \pm 0.18$ | $3.16 \pm 2.03$ | $-0.03 \pm 0.82$ | $2.9 \pm 0.63$ |
| **FC** | $89.9 \pm 3.60$ | $0.16 \pm 0.07$ | $0.38 \pm 0.17$ | $5.35 \pm 2.50$ | $1.5 \pm 5.87$ | $18.1 \pm 5.90$ |
| **TC** | $85.6 \pm 4.54$ | $0.23 \pm 0.12$ | $0.60 \pm 0.38$ | $6.35 \pm 4.36$ | $-1.0 \pm 11.92$ | $24.9 \pm 6.79$ |

method was evaluated using datasets from the SKI10 challenge as well as from the OAI database. Accuracy was evaluated using volume-based and distance-based measures to provide a transparent analysis w.r.t global and local level of detail. The proposed method consistently achieved high segmentation accuracy, despite severely arthritic knees and various different MRI sequences. For the first time, a total score greater than 75 was reached on the *SKI10* submission data, which is comparable to the inter-observer variability of two expert readers [11]. In our experience, the automated method reduces the time effort for an accurate segmentation of knee bones and cartilage at least by a factor of six compared to manual segmentations by an experienced reader (>1h). However, large scale databases for studying the OA disease, such as provided by the OAI, can contain 50.000 or more MRIs. Using our implementation, it would take 43 weeks to segment the full OAI database on a singe computational node. We therefore aim at reducing the computational time of the algorithm further as well as distributing the work-load over several nodes. Our goal is to segment the full OAI database and make the results available to the public in the near future.

A promising line of future work is to investigate approaches that couple SSMs and CNNs more directly, e.g. by introducing learned appearance from CNNs to an SSM segmentation framework.

**Acknowledgments**

We would like to thank Heiko Ramm (née Seim, 1000shapes GmbH) for valuable insights into SSM-based segmentation methods. We further would like to thank Irene Ziska, Agnieszka Putyra, and Robert Joachimsky for creating the manual segmentations for our *OAI ZIB* dataset by thoroughly correcting automated presegmentations. The authors gratefully acknowledge the financial support by the German federal ministry of education and research (BMBF) research network on musculoskeletal diseases, grant no. 01EC1408B (Overload/PrevOP) and grant no. 01EC1406E (TOKMIS). The Osteoarthritis Initiative is a public-private partnership comprised of five contracts (N01-AR-2-2258; N01-AR-2-2259; N01-AR-2-2260; N01-AR-2-2261; N01-AR-2-2262) funded by the National Institutes of Health, a branch of the Department of Health and Human Services, and conducted by the OAI Study Investigators. Private funding partners include Merck Research Laboratories; Novartis Pharmaceuticals Corporation, GlaxoSmithKline; and Pfizer, Inc. Private sector funding for the OAI is managed by the Foundation for the National Institutes of Health. This manuscript was prepared using an OAI public use data set and does not necessarily reflect the opinions or views of the OAI investigators, the NIH, or the private funding partners.

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

## Appendix A: Employed surface-based and volume-based distance measures

$$\text{DSC} = 100 \cdot \frac{2 \mid B \cap A \mid}{\mid B \mid + \mid A \mid} \, , \; \text{VOE} = 100 \cdot \left(1 - \frac{\text{DSC}}{200 - \text{DSC}}\right) , \; \text{VD} = 100 \cdot \frac{\mid B \mid - \mid A \mid}{\mid A \mid}, \quad (3)$$

$$\text{ASD} = \frac{1}{n_{\partial A} + n_{\partial B}} \left( \sum_{i=1}^{n_{\partial A}} \min_{b \in \partial B} \|a_i - b\|_2 + \sum_{j=1}^{n_{\partial B}} \min_{a \in \partial A} \|b_j - a\|_2 \right),$$

$$\text{RSD} = \sqrt{\frac{1}{n_{\partial A} + n_{\partial B}} \left( \sum_{i=1}^{n_{\partial A}} \min_{b \in \partial B} \|a_i - b\|_2^2 + \sum_{j=1}^{n_{\partial B}} \min_{a \in \partial A} \|b_j - a\|_2^2 \right)}, \quad (4)$$

$$\text{MSD} = \max \left( \max_{a \in \partial A} \min_{b \in \partial B} \|a - b\|_2, \; \max_{b \in \partial B} \min_{a \in \partial A} \|b - a\|_2 \right).$$

Within the above, $A$ denotes the set of manually segmented (ground-truth) voxels and $B$ denotes the segmentation result from the automated method; $\partial A$ and $\partial B$ represent the boundary of $A$ and $B$. The boundary contains every voxel having at least one neighbor that is not part of the respective segmentation mask. The number of voxels on the boundary $\partial A, \partial B$ is written as $n_{\partial A}, n_{\partial B}$. Lastly $\mid \cdot \mid$ denotes a volume and $\|\cdot\|_2$ the usual Euclidean norm.

## Appendix B: *SSM postprocessing* schematic visualization

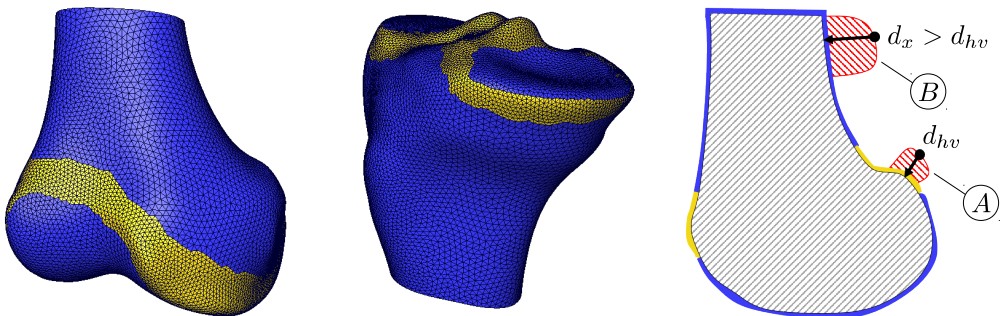

Figure 5: Predefined regions of high (yellow) and low (blue) variability on SSM surfaces of FB (left) and TB (middle). Right: Schematic visualization of a typical postprocessing situation. Distances are computed from 3D CNN mask (red) to SSM surface. The cross-section depicted shows (w.l.o.g.) the realization of $d_{hv}$ in component $A$. Component $B$ is completely removed since there exists one $x \in B$ s.t. $d_x > d_{hv}$ and all elements of $B$ realize their closest distance in the area of low variability.

## Appendix C: Sampling for *CNN-3D* – visualization

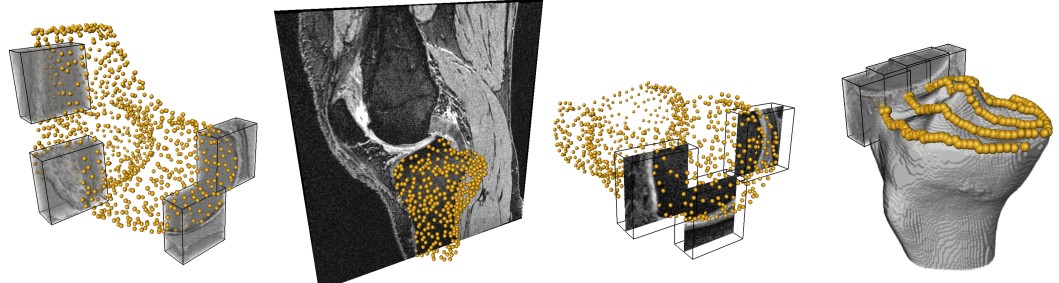

Figure 6: *CNN-3D* segmentation step: OAI subject ID 9793168. Exemplary sampling of subvolumes for (f.l.t.r.) FB (1052 points), TB (825 points), FC (845 points) and TC (228 points).

