# OpenReview forum: "Automated Segmentation of Knee Bone and Cartilage combining Statistical Shape Knowledge and Convolutional Neural Networks: Data from the Osteoarthritis Initiative"
_MIDL.amsterdam/2018/Conference — MIDL 2018 Oral_

### Review · AnonReviewer3 · 2018-04-30
**Review of Automated Segmentation of Knee Bone and Cartilage using Statistical Shape and CNNs**

**Rating:** 5
**Confidence:** 2

**Review:**

This paper introduces a new approach for the segmentation of the femoral bone (FB), tibial bone (TB), and the respective femoral and tibial cartilage (FC,TC) from MRIs.  The technique proposed for such task is based on a combination of statistical shape model (SSM) and deep learning, involving the following steps: 1) slice by slice 2-D FB and TB segmentation using a 2-D U-net (CNN) trained with a Dice loss, where the input contains 8 slices above and below the slice to be segmented (they serve as a context information for the segmentation); 2) SSM shape adjustment in 3-D that regularizes the segmentation from step (1) above; 3) sub-volume FB and TB segmentation using a 3-D U-net (CNN) trained with a Dice loss, where the results from the sub-volume segmentation are combined via majority voting in overlapping regions (i.e., volume regions that have more than one sub-volume segmented); 4) SSM-Postprocessing that regularises the FB and TB  segmentation from step (3) by removing wrongly classified foreground labels located outside the typical range of osteophytic growth; and 5) FC and TC segmentation using 3D U-net.  Experiments based on volume-based (Dice, volume difference, volume overlap error) and distance-based (average surface distance, root mean square distance, maximum distance) measures are shown on three datasets: SKI10, OAI Imorphics and OAI ZIB.  On SKI10, the proposed approach achieves the currently best results in the field.  Results on OAI Imorphics and OAI ZIB are harder to interpret since there is no comparison with any baseline, and OAI ZIB appears to be not so challenging.

I think this is a solid paper with interesting results and enough technical novelty, so I recommend it to be accepted for MIDL.  My main recommendation is for the authors to address one of the issues mentioned in Section 4: how to couple SSMs and CNNs more directly - in particular how could a joint SSM-CNN model be trained end-to-end?



**Special Issue:**

Yes

---

> ### Comment · ~Felix_Ambellan1 · 2018-05-14
> **Re: Review of Automated Segmentation of Knee Bone and Cartilage using Statistical Shape and CNNs**
>
> Thank you for your helpful comments.
>
> We would like to address your main issue:
>
> >>Results on OAI Imorphics and OAI ZIB are harder to interpret since there is no comparison with any baseline, and OAI ZIB appears to be not so challenging.<<
>
> We refer here to our answer given to reviewer 'AnonReviewer1'. The mentioned comment addresses the baseline issue for OAI-Imorphics and OAI-ZIB in details. Furthermore, OAI-ZIB contains 60.9% patients with an OA grade of 3 and 4, thus they feature (heavy) ostheophytic growth. This ostheophytic growth is strongly patient specific and due to its individuality pretty challenging for any segmentation approach. The OAI-ZIB data additionally contains several cases with cartilage denudations, which also pose a challenge for segmentation.
>
> >>My main recommendation is for the authors to address one of the issues mentioned in Section 4: how to couple SSMs and CNNs more directly - in particular how could a joint SSM-CNN model be trained end-to-end?<<
>
> In case our paper is considered for submission to the Special Issue we could also focus more on the possibilities of coupling SSMs and CNNs in the discussion. In the current setting this seems to be impractical, since we would probably have to overexpand the format frame of eight pages.

---

### Review · AnonReviewer2 · 2018-05-08
**Novel U-Net based segmentation pipeline that uses SSMs for regularisation and post-processing**

**Rating:** 5
**Confidence:** 2

**Review:**


Summary:
A novel segmentation pipeline comprising of 2D and 3D U-Nets along with a-priori statistical shape models (SSM)  are used to segment knee bone and cartilage from MRI. The method uses SSMs for regularisation and post-processing to improve the segmentation results. The method currently ranks as the best performing method in the MICCAI SKI10 challenge, and is also evaluated on two other MRI datasets yielding competitive results.

Pros

1. Thorough literature survey is presented relating to SSM and CNN based segmentation methods.
2. Incorporating a-priori SSMs is a novel and interesting contribution when using the two step 2D and 3D U-Net based segmentation. This can be of interest in other medical image applications where shape models are available.
3. Extensive evaluation of the method on three datasets
4. Performance on the MICCAI SKI10 challenge is impressive.
5. The paper is clearly written.

Cons

1. Pointers to deep learning based methods that use some form of shape models could be useful to position the current work such as [1]. This also relates to the discussion on how to couple SSMs and CNNs in Section 4.
2. Use of SSMs obtained from SKI10 is used on OAI Imorphics in Section 3.2, which is a strong point if it is transferable but discussion on what the impact of this is missing. If SSMs are unavailable for one dataset, to what degree are SSMs from other datasets useful?

References

[1] Oktay, Ozan, et al. "Anatomically constrained neural networks (ACNNs): application to cardiac image enhancement and segmentation." IEEE transactions on medical imaging 37.2 (2018): 384-395.
https://arxiv.org/pdf/1705.08302.pdf

**Special Issue:**

Yes

---

> ### Comment · ~Alexander_Tack1 · 2018-05-14
> **Re: Novel U-Net based segmentation pipeline that uses SSMs for regularisation and post-processing**
>
> Thank you for your valuable remarks.
>
> We would like to address the mentioned cons:
>
> 1. Thank you for providing us this very recently published paper. If possible in a revised final article, we will gladly include this paper and maybe additional ones to illustrate the use of shape knowledge in combination with deep learning.
>
> 2. We agree. It is indeed a strong point, that SSMs not generated using the data of a specific cohort can be applied for the task of segmentation of these data. We will further highlight this strength of our method in a revised version in case of acceptance.

---

### Review · AnonReviewer1 · 2018-05-09
**A good paper**

**Rating:** 3
**Confidence:** 3

**Review:**

Overall:
The paper proposes a solution to the problem of knee bone and cartilage segmentation from MRI using deep learning (2D and 3D U-Nets) and Statistical Shape Knowledge (SSM). In general the idea is sound and it received state-of-the-art results in the MICCAI SKI10 challenge. My main concern is whether SSM components allow to learn CNN through back-propagation. Except that, the paper is very interesting and it is a solid contribution to the conference.

Strengths:
+ The proposed approach is interesting solution to the problem of knee bone and cartilage segmentation from MRI.
+ The method combines a deep learning approach with Statistical Shape Knowledge (SSM).
+ The results are impressive since the proposed approach obtained the best average performance in the MICCAI SKI10 grand challenge.

Remarks:
* Major
- I have big doubts whether SSM-Adjustment and a voting mask allow for gradient flow. In other words, whether these two components allow to learn CNNs through back-propagation. What is a learning procedure in the considered case?
- What is the point of presenting results for OAI datasets if there is no baseline we can compare to?

* Minor
- Why names of the last two authors are not in bold?
- The paper needs additional editing since there are missing spaces between paragraphs.
- The authors state that a single segmentation takes ~9.5min. How slow is the application of SSM comparing to the CNN-based components? It seems that SSM is a bottleneck and this could be easily improved in the future.

**Special Issue:**

No

---

> ### Comment · ~Felix_Ambellan1 · 2018-05-14
> **Re: A good paper**
>
> Thank you for pointing out two major issues:
>
> 1. Connection of CNN and SSM.
>
> Within our segmentation pipeline there is currently no coupling between CNNs and SSMs. This remains, as stated in the conclusion, as open question for future work. There is nothing like an SSM-Adjustment or voting mask layer and we agree that it might be a problem to create something like that. Nevertheless, one option to connect CNNs snd SSMs could be i.e. learning of an appearance model for the SSM deformation by the means of an CNN.
>
> 2. Missing baseline for the OAI datasets.
>
> At this point we did not clearly state that there is a result of another group for parts of the OAI-Imorphics dataset that we outperform by approximately 5% w.r.t. dice similarity.
>
> In details:
> Dam et al. (cited as no. 14 in the paper) received a dice similarity of 81.2 +- 5.5% (MTC) and 86.6 +- 3.4% (LTC) for the baseline data. Whereas we achieve 86.1 +-5.3% (MTC) and 90.4 +- 2.4% (LTC).  They also provide numbers for the femoral cartilage, but they splitted the cartilage interface w.r.t. the two condyles into Medial Femoral Cartilage (MFC) and Lateral Femoral Cartilage (LFC), s.t. their numbers are not directly comparable to ours. They receive 81.4 +-4.4% (MFC) and 84.2 +- 4.3%, whereas we achieve 89.4 +- 2.4% for the complete femoral cartilage interface.
>
> For the OAI-ZIB dataset there are indeed no results of other groups since we created it for this study. The main goal of using it is to show that the proposed method works on several different datasets of different scales (cohort size) and  different image modalities. We wanted to point out that the method's performance does not depend heavily on the cohort size. The fact that the results of OAI-ZIB, OAI-Imorphics and SKI10 provide a comparable level of accuracy helps to further put trust into the ground truth of our readers.
>
> Thank you as well for your minor remarks. We will acknowledge them in a revised version in case of acceptance.

---

### Comment · ~Bram_van_Ginneken1 · 2018-05-18
**Selection for longlist for special issue Medical Image Analysis**

Dear authors,

Congratulations on your acceptance to MIDL! We have selected your paper on the longlist for the Medical Image Analysis Special Issue. Please read this page:
https://midl.amsterdam/special-issue-in-medical-image-analysis/
Please answer the three questions that are listed on that page about your interest in submitting to the special issue, potential overlap with other publications, and related publications.

You can post your answer here directly below on openreview.net, or mail me directly at bram.vanginneken@radboudumc.nl.

Best regards, Bram

---

> ### Comment · ~Felix_Ambellan1 · 2018-05-22
> **Re Selection for longlist for special issue Medical Image Analysis**
>
> Hello Bram,
>
> here is what we want to respond to your previous mail concerning the special issue long list:
>
> 1. To indicate if they are interested to be eligible for the special issue. This would mean they are willing to augment the content of their paper significantly and submit the full manuscript before August 1, 2018.
>
> We are glad to be considered for the special issue and are also willing to prepare a manuscript.
>
>
> 2. To confirm that the paper, or any paper with overlap with the contents of the MIDL paper, is and will not be under review or under consideration elsewhere, following the Elsevier rules on “Multiple, redundant or concurrent publication” (see here: https://www.elsevier.com/authors/journal-authors/policies-and-ethics).
>
> Neither the MIDL paper itself, nor any other paper with overlapping content is under consideration elsewhere.
>
>
> 3. To send us any related publications of the author group, for example, other papers that apply a similar methodological idea to another data set, to help reviewers judge the novel contribution of the work.
>
> There is one publication to mention:
>
> A.Tack, A. Mukhopadhyay, S.Zachow; Knee menisci segmentation using convolutional neural networks: data from the Osteoarthritis Initiative; Osteoarthritis and Cartilage: Volume 26(5), pp 680-688, May 2018
>
> This paper was written simultaneously to the MIDL paper, with a strong focus on the osteoarthritis-related clinical biomarkers (e.g. meniscal extrusion).
> In our MIDL submission we, in contrast, focus on the methodological description, work with different structures of the knee, on multiple MRI sequences, and provide a thoroughly evaluation on large datasets as well as state-of-the-art results for the SKI10 challenge.
>
> Best regards,
>
> Alex, Moritz, Stefan and Felix

---

### Decision · Program_Chairs · 2018-05-15
**Paper75 Acceptance Decision**

Oral